# Training on Sand or Parquet: Impact of Pre-Season Training on Jumping, Sprinting, and Change of Direction Performance in Professional Basketball Players

Jo-Lâm Vuong *, Johanna Heil, Nina Breuer, Matthaios Theodoropoulos, Nicola Volk [ID], Antonia Edel and Alexander Ferrauti

Department of Training and Exercise Science, Faculty of Sport Science, Ruhr University Bochum, 44801 Bochum, Germany
* Correspondence: jo-lam.vuong@ruhr-uni-bochum.de

**Abstract:** This study examined the effects of combined change of direction (CoD) and plyometric training on sand in basketball players' jump, sprint, and CoD performances. In total, twenty-five male professional basketball players (age, $24.1 \pm 4.2$ years; height, $192.5 \pm 8.0$ cm; body mass, $92.6 \pm 11.6$ kg) participated in a pre-season intervention study over seven weeks, where two teams completed an identical speed training protocol, either on sand or on a hard surface, while a third team served as the control. All participants followed their regular pre-season training, while the intervention groups additionally performed the training protocol twice weekly. A repeated-measures ANOVA (rANOVA) showed significant interaction effects on the countermovement jump ($F = 14.90$, $p < 0.001$), pivot–step–jump test ($F = 8.09$, $p = 0.002$), 10 m sprint time (ST) ($F = 4.18$, $p = 0.050$), and 20 m ST ($F = 8.49$, $p = 0.002$). Moreover, an rANOVA showed significant interaction effects for the CoD performance regarding total time ($F = 5.70$, $p = 0.010$) and average velocity prior to the CoD ($F = 8.40$, $p = 0.002$) and after the CoD ($F = 3.89$, $p = 0.036$). As such, the findings suggest that sand-based training elicits kinematic adaptations, increased muscle activation, and a shift towards concentric force development that all contribute to enhanced athletic capabilities.

**Keywords:** change of direction; agility; speed training; sand training



## 1. Introduction

In recent decades, technical innovations, rule changes, and a greater focus on the physical development of athletes have resulted in team sports becoming increasingly faster [1–3]. Especially in modern high-level basketball, drastic rule changes (shortened offensive time, less time to bring the ball over the midline, division of playing time into shorter quarters) have turned the game into one of the fastest team sports and altered the demands placed on players accordingly [4,5]. In line with these changes, recent studies show an increase in the frequency of high-intensity actions, such as decelerations, accelerations, changes of direction (CoDs), and jumps, compared to earlier findings [6–9]. Accordingly, the results of some studies show that high-level players differ from lower-level players in their speed, particularly their CoD speed [10,11].

In many cases, directional changes occur in response to an external stimulus [12,13]. Thus, on the court, CoD speed manifests itself mostly in combination with the cognitive processes of perception and decision making. This complex interaction of cognitive and motor performance has recently been termed agility, and it is likely that CoD speed, as a physical and technical component, is an underlying quality of the more complex agility. Therefore, CoD speed is the result of the interplay between different conditional and coordinative factors, and it involves the ability to decelerate running speed in the shortest possible time, reorient the body, align it accordingly, and accelerate as quickly as possible in a new direction [14,15].

In response to these findings, scientists and strength and conditioning practitioners are continually searching for effective training approaches to optimize CoD speed, various examples of which (strength training, explosive strength training, plyometrics, sprint training, deceleration, acceleration, and CoD drills) have been proven effective [13,16–19]. Some recent studies have also suggested the use of sand as an effective alternative training surface to improve jumping and sprinting [19–22]. Sand as a training surface potentially provides a natural tool for increasing resistance to movement, as it induces adaptations at the physiological, mechanical, and neuromuscular levels [23,24]. The instability and force-absorbing properties of sand require heightened stabilization, elongated ground contact time, and increased muscular exertion [25] This makes sand training a potentially valuable tool for improving speed, especially CoD speed.

Since speed and CoD training is usually associated with high loads on the musculoskeletal system, sand-based training might be a way to achieve this with reduced impact. Several studies have already demonstrated a positive effect on the speed performance of plyometric training on sand [19,23,26,27], but to the best of our knowledge, no previous study has investigated the effects of combining CoD speed training specifically with plyometrics on a sand surface. Therefore, the objective of this study is to evaluate the effectiveness of sand as a training surface for enhancing jump performance, sprinting ability, and CoD speed on hard surfaces. This study focuses on assessing and comparing the effects of a combined training protocol consisting of plyometric exercises and CoD speed drills on a sand surface versus a hard surface, with a specific emphasis on evaluating CoD performance alongside jump and sprint capabilities.

## 2. Materials and Methods

### 2.1. Participants

In total, twenty-five professional basketball players (age, 24.6 ± 4.3 years; height, 192.5 ± 8.2 cm; body mass, 92.6 ± 11.8 kg) from three teams competing in the second German national basketball league participated in this study. The anthropometric and demographic data split by intervention group are presented in Table 1. All participants were elite-level athletes and, thus, highly experienced in terms of physical training. The training load in terms of basketball, strength, and conditioning sessions among the three teams was controlled to be comparable over the intervention period, though one team completed a training intervention on sand (SG), a second completed the same intervention on a hard surface (HG), and a third team was designated the control group (CG). Before this study began, all subjects were informed in detail about possible risks and the data privacy policy, as well as that each participant could terminate their participation at any time without providing a reason. Written informed consent was obtained from the subjects, and the Ethics Committee of the Faculty of Sports Science of the Ruhr University Bochum (EKS V 11/2022) approved the study design, procedures, and measurements, which are also in line with the Declaration of Helsinki.

**Table 1.** Descriptive data of anthropometric and demographic parameters for the surface conditions and control group (*n* = 25).

| Condition | Age [Years] | Weight [kg] | Height [cm] |
|:---:|:---:|:---:|:---:|
| SG | 26.7 ± 2.8 | 98.7 ± 14.0 | 194.3 ± 7.9 |
| HG | 24.2 ± 4.6 | 88.5 ± 9.7 | 189.3 ± 8.7 |
| CG | 22.6 ± 4.9 | 90.3 ± 9.3 | 194.4 ± 7.6 |

Values are given as the mean ± standard deviation (SD); SG = sand group; HG = hard surface group; CG = control group.

### 2.2. Procedures

The training interventions were conducted over a period of seven weeks, one on a sand surface with a depth of 40–45 cm and a balanced mixing ratio of fine and coarse grains (0.1–1.2 mm; DIN EN 1177) and the other on elastic beech parquet with a swing beam

construction as a substructure. One week prior to the start of the intervention and one week after the intervention, various performance tests were carried out. In the sixth week, the acute effects of a single training session were also assessed among the intervention groups ($n = 18$), which completed two training sessions weekly on sand (SG; $n = 9$) or a hard surface (HG; $n = 9$) for a period of 7 weeks, with an interval of 3–4 days between sessions, which lasted 30–40 min. A third group (CG; $n = 7$) did not complete any additional training sessions. After the seventh week, a taper week was held before subsequent performance testing was carried out. The same trained instructors led all training sessions, and each participant had to complete at least 12 sessions over the intervention period for the data to be included in the study analysis.

The training protocol consisted of various CoD exercises in combination with different plyometrics. The protocol for both intervention groups was identical, but all jumps performed on sand rely on the long stretch-shortening cycle (SSC) due to the prolonged ground contact times caused by the properties of sand, whereas some jumps rely on the short SSC when performed on hard surfaces. To ensure variety and prevent monotony, certain exercises were modified weekly in terms of movement patterns or the number of directional changes. The training protocol followed a weekly progression regarding the CoDs to be completed, the total running distance, and the number and intensity of plyometrics. Further, the training volume was designed according to studies that have conducted similar interventions on hard surfaces [28–30]. To avoid fatigue-induced performance decrements, recovery intervals of 30 s within each set and 2 min between sets were maintained. The exact training protocol can be seen in the appendix (Figures A1 and A2).

### 2.3. Measurements

All participants underwent a series of performance tests to evaluate their linear acceleration, horizontal and vertical jumping power, and CoD speed. These tests were conducted one week before and one week after the intervention. The teams' performance assessments took place in their training facilities, with all participants being tested on parquet surfaces. These assessments were consistently scheduled in the early evening. Since the tests were conducted indoors in facilities that meet the league's standards, the environmental conditions were identical. The test battery, which required two hours to complete, was administered exclusively by trained test administrators. In addition, heart rate (HR), blood lactate (LA), rate of perceived exertion (RPE), and counter movement jump (CMJ) were assessed in one session to evaluate the acute effects of sand-based training.

### 2.3.1. Anthropometrics

Anthropometric measurements comprised body mass and height, the former of which was measured with a digital scale (ADE Electronic Column Scales, Hamburg, Germany, measurement accuracy ±0.1 kg) and the latter with a fixed stadiometer, which meets the standards of the International Society for the Advancement of Kinanthropometry (ISAK, Holtain Ltd., Crymych, UK, measurement accuracy ±0.1 cm). All measurements were performed by trained test supervisors in accordance with the ISAK guidelines [31].

### 2.3.2. Heart Rate

HR was measured at rest and immediately after the completion of one single training session using Acentas' team software and the corresponding chest belts (Acentas GmbH, Hörgertshausen, Germany, measurement accuracy ±1 bpm).

### 2.3.3. Blood Lactate Concentration

LA concentration was determined from capillary blood samples using enzymatic amperometry. Blood samples were taken using 20 μL capillary tubes from the earlobe at rest and immediately after the last drill was finished. The samples were hemolyzed in 1 mL micro-test tubes and analyzed for LA using a Biosen S-Line Lab+ (EKF-Diagnostik 141 GmbH, Magdeburg, Germany) in the laboratory within 24 h of testing.

### 2.3.4. Rate of Perceived Exertion

The subjective perception of exertion was measured using the standardized RPE scale, ranging from 1 to 10, prior to and following the completion of one single training session.

### 2.3.5. Countermovement Jump

Vertical jump height [cm] for the CMJ was calculated using the time-of-flight method $[h = \frac{1}{2} \times g \times \left(\frac{t}{2}\right)^2]$, where flight time was measured using a contact mat (Haynl Elektronik, Schönebeck, Germany, measurement accuracy 1000 Hz). The subjects were positioned on the mat with their hands on their hips in a parallel stance. From this position, the players bent their hips and knees to a self-selected depth and performed a maximum vertical jump out of this countermovement without using the arms [32]. Each participant completed three attempts, with a pause time of 45 s between jumps.

### 2.3.6. Repetition Jumps

The jump heights [cm] and contact times [s] of the repetition jumps (RJ) performed were recorded using a contact mat (Haynl Elektronik, Germany, measurement accuracy 1000 Hz). The subjects were instructed to jump with their hands on their hips such that they had minimum ground contact time while jumping to their maximum height. Each subject performed 10 consecutive jumps, of which the 3 with the highest jump efficiency score defined by the reactive strength index (RSI) were analyzed. For the analysis, the RSI was calculated from the mean value of these three jumps as the ratio of jump height (JH) to contact time (CT) ($RSI = \frac{JH}{CT}$) [33].

### 2.3.7. Pivot–Step–Jump Test

The jump height [cm] of the pivot–step–jump test (PSJT) was recorded using a contact mat (Haynl Elektronik, Germany, measurement accuracy 1000 Hz). Subjects were positioned at a 90-degree angle to the contact mat so they could reach it with one short step. When starting the jump, they performed a 90-degree forward pivot step onto the mat, followed by a double-legged vertical jump, including the use of the arms [34].

### 2.3.8. Crossover Hop for Distance

Horizontal jump distance [cm] was determined with the crossover hop for distance (CH), where subjects initially stood on the designated test leg with their toes on the starting line. If subjects jumped with their right leg, they stood on the right side of the measuring tape attached to the floor and vice versa. Subjects were instructed to complete three consecutive maximum horizontal jumps, crossing the measuring tape each time, and distance was measured from the starting line to the point where the subject's heel landed after the third jump [35]. The subjects completed two attempts with each leg, of which the best one was evaluated. Between attempts, a recovery time of 45 s was maintained.

### 2.3.9. Sprint Times

Linear sprint times (ST) [s] were recorded for distances of 5, 10, and 20 m using a photoelectronic double light gate system (Witty System, Microgate, Bolzano, Italy, measurement accuracy ±0.001 s). The participants were instructed to begin at a starting line placed on the floor 50 cm in front of the first light gate, with both soles of their feet on the ground. All subjects completed two maximum sprints, the best of which was evaluated. A 3 min passive recovery period was maintained between attempts to prevent a fatigue-related performance decrease [36].

### 2.3.10. Modified 5-0-5 Test

The subjects' performance [s, m/s, m] of a 180° CoD was assessed using a modified 5-0-5 test. The data for deceleration, acceleration, and total time (TT) were recorded using a motorized resistance device (1080 Motion Sprint, Lidingö, Sweden, measurement accuracy 333 Hz) placed 3 m behind the starting line, and a towing cable was attached to the hip of

the subject using a pear-shaped carabiner and a tightly laced sling rope. The tightening knot was placed on the contralateral side of the turning foot to allow an undisturbed swivel of the carabiner [37], and a resistance of 3 kg was applied over the entire test. The participants were instructed to begin at a starting line placed on the floor with both soles of their feet on the ground. From this position, they maximally accelerated toward a 5 m distant target marker, which had to be crossed with both feet before they executed a 180° CoD and accelerated back as quickly as possible over the starting line. Two attempts, each pivoting on both the left and right feet, were completed, of which the best time was recorded to the nearest 0.01 s [38–40]. The device was used to measure the total time required (TT), average velocity before (Avg Vel 1a) and after the CoD maneuver (Avg Vel 1b), as well as the deceleration distance (Dec Dist), starting with the first decrease in velocity.

*2.4. Statistical Analysis*

The Shapiro–Wilk test was used to assess the normality of the data, and a significance level of $p \leq 0.05$ was employed to determine whether the data followed a normal distribution. To investigate potential group differences in the acute effects of a specific training session on metabolic demands, perceived exhaustion, and jumping performance, a two-factor (time × intervention) repeated-measures analysis of variance (rANOVA) was carried out, employing a significance level of $p \leq 0.05$. Similarly, two-factor (time × intervention) rANOVAs were conducted to assess the effects of the intervention on the jump, sprint, and CoD performance with the same significance level. The $\alpha$-level was adjusted using Tukey's correction, statistics of significant post-hoc tests are provided as a range of the mean difference (MD), and the smallest lower and highest upper 95% confidence limits (CLs) are displayed in square brackets as [smallest lower; highest upper].

Moreover, the data were assessed by analyzing practical relevance using magnitude-based inferences (MBIs), calculated starting from the 90% CL using a published spreadsheet [41] to assess the likelihood that a change in average is practically relevant [42]. To determine the threshold values, the smallest worthwhile change (SWC) was calculated for a small effect (0.2). To calculate this, the between-subject standard deviation of each performance variable was multiplied by 0.2 ($SWC = 0.2 \times SD$). For instance, if an athlete jumps 45 cm in the CMJ with a standard deviation of 4 for this test in their population, the athlete would have to jump 0.8 m higher to achieve a meaningful change. A CoD was categorized as follows based on the probability that the true value of the standardized mean difference (SMD) will be greater than the SWC: (1) "Possibly" was assigned when the true value of the SMD had a probability ranging from 25% to 75% of being greater than the SWC; (2) "Likely" was assigned when the true value of the SMD had a probability ranging from 75% to 95% of being greater than the SWC; and (3) "Very likely" was assigned when the true value of the SMD had a probability ranging from 95% to 99.9% of being greater than the SWC [42].

In addition, a Pearson correlation was calculated with the total sample in order to examine correlations between the performance developments. For this purpose, the differences between pre and post values were calculated, and a significance level of $p \leq 0.05$ was applied. Statistical analyses of the data were performed using the SPSS analysis software (version 27.0; SPSS Inc., Chicago, IL, USA), Microsoft Excel (version 16.16.5; Microsoft Corp., Redmond, WA, USA), and Jamovi (version 2.2.5.0).

## 3. Results

A comparison of the acute effects of one training session between the SG and the HG was performed using an rANOVA, which showed significant time × intervention interaction effects for the following variables: CMJ (F = 4.68, $p = 0.038$); LA (F = 11.0, $p = 0.003$); and session RPE (F = 20.0, $p < 0.001$). The results are presented in Table 2.

**Table 2.** Comparison of the SG (*n* = 9) and the HG (*n* = 9) in terms of the acute effects of a single training session.

|  |  | Pre-Session | Post-Session | rANOVA |
|---|---|---|---|---|
| CMJ [cm] | SG | 34.9 ± 4.8 | 40.9 ± 5.7 a | *p* = 0.038 |
|  | HG | 38.2 ± 5.4 | 40.7 ± 5.8 |  |
| LA [mmol/L] | SG | 0.9 ± 0.2 | 3.4 ± 1.8 a,b | *p* = 0.003 |
|  | HG | 0.8 ± 0.2 | 1.5 ± 0.7 |  |
| HR [bpm] | SG | 77.9 ± 10.0 | 150.0 ± 15.0 | *p* = 0.072 |
|  | HG | 88.6 ± 33.6 | 141.0 ± 21.9 |  |
| RPE | SG | 1.6 ± 0.5 | 7.7 ± 0.7 a,b | *p* < 0.001 |
|  | HG | 1.4 ± 0.5 | 5.2 ± 1.2 |  |

Values are given as mean ± SD; a = sig. dif. to pre, b = sig. dif. SG–HG; pre-session = values were evaluated before the start of the training session, post-session = values were evaluated right after the training session was finished; SG = sand group, HG = hard court group, CMJ = countermovement jump, RPE = rate of perceived exertion on a scale of 1 to 10.

A post-hoc pairwise comparison analysis revealed that the SG performed better in the CMJ after the training session compared to their jumping height before (MD [95% CL] = 0.16–6.01 [31.0–45.1] cm, $p^{Tukey}$: <0.001). Following the training session, the SG showed a substantial elevation in LA, accompanied by a statistically significant distinction when compared with the post-session LA values in the HG (MD [95% CL] = 0.16–2.64 [0.69–4.31] mmol/L, $p^{Tukey}$: <0.001–0.005). In terms of session RPE, the rating numbers of both the SG and HG increased after the session, with significantly higher ratings for the SG compared to the HG (MD [95% CL] = 0.19–6.31 [1.17–8.52] RPE, $p^{Tukey}$: <0.001). No differences were found in HR.

The analysis of longitudinal training effects on jumping and sprint performance parameters among the SG, HG, and CG was conducted using an rANOVA, which revealed no group differences pre-intervention, but significant time × intervention interaction effects for the following variables: CMJ (F = 14.90, *p* < 0.001); PSJT (F = 8.09, *p* = 0.002); 10 m ST (F = 4.18, *p* = 0.051); and 20 m ST (F = 8.49, *p* = 0.002).

A post-hoc pairwise comparison analysis revealed that only the SG improved significantly throughout the intervention period in CMJ (MD [95% CL] = 0.03–4.88 [39.3; 49.0] cm, $p^{Tukey}$: <0.001); PSJT (MD [95% CL] = 0.29–5.14 [50.3; 63.6] cm, $p^{Tukey}$: <0.001); 10 m ST (MD [95% CL] = 0.00–0.13 [1.70; 1.91] s, $p^{Tukey}$: 0.018–0.032); and 20 m ST (MD [95% CL] = 0.01–0.91 [2.86; 3.22] s, $p^{Tukey}$: <0.001–0.050) performance.

Moreover, the analysis of magnitude-based inferences showed that assuming a small effect, the changes in CMJ performance were highly likely beneficial for both the SG and the HG. This was also true for PSJT, CH, 5 m ST, 10 m ST, and 20 m ST performance, but only in the SG. Concerning RSI performance, the analysis showed that the change was likely beneficial for the HG. Statistical values for jump and speed performances are shown in Table 3, and the individual values of and the percentage changes in the RSI, CMJ, and 10 m ST are presented as examples in Figure 1.

To analyze the longitudinal training effects of the intervention on the parameters of CoD performance among the SG, HG, and CG, an rANOVA was performed. The analysis revealed no group differences pre-intervention but significant time × intervention interaction effects for the following variables: TT (F = 5.70, *p* = 0.010); Avg Vel 1a (average velocity from start to CoD) (F = 8.40, *p* = 0.002); and Avg Vel 1b (average velocity after CoD) (F = 3.89, *p* = 0.036).

A post-hoc pairwise comparison analysis revealed improvements throughout the intervention in TT in both the SG and HG (MD [95% CL] = 0.01–0.20 [2.79; 3.20] s, $p^{Tukey}$: 0.019–0.042). Moreover, an increase in Avg Vel 1a was observed for the HG (MD [95% CL] = 0.01–0.21 [2.75; 3.22] m/s, $p^{Tukey}$: 0.016), whereas the SG improved in Avg Vel 1b and also exhibited a difference between the SG and CG post-intervention

(MD [95% CL] = 0.01–0.16 [3.41; 3.74] m/s, $p^{\text{Tukey}}$: 0.008–0.049). No significant differences were found in Dec Dist (distance from the first drop in velocity to the CoD).

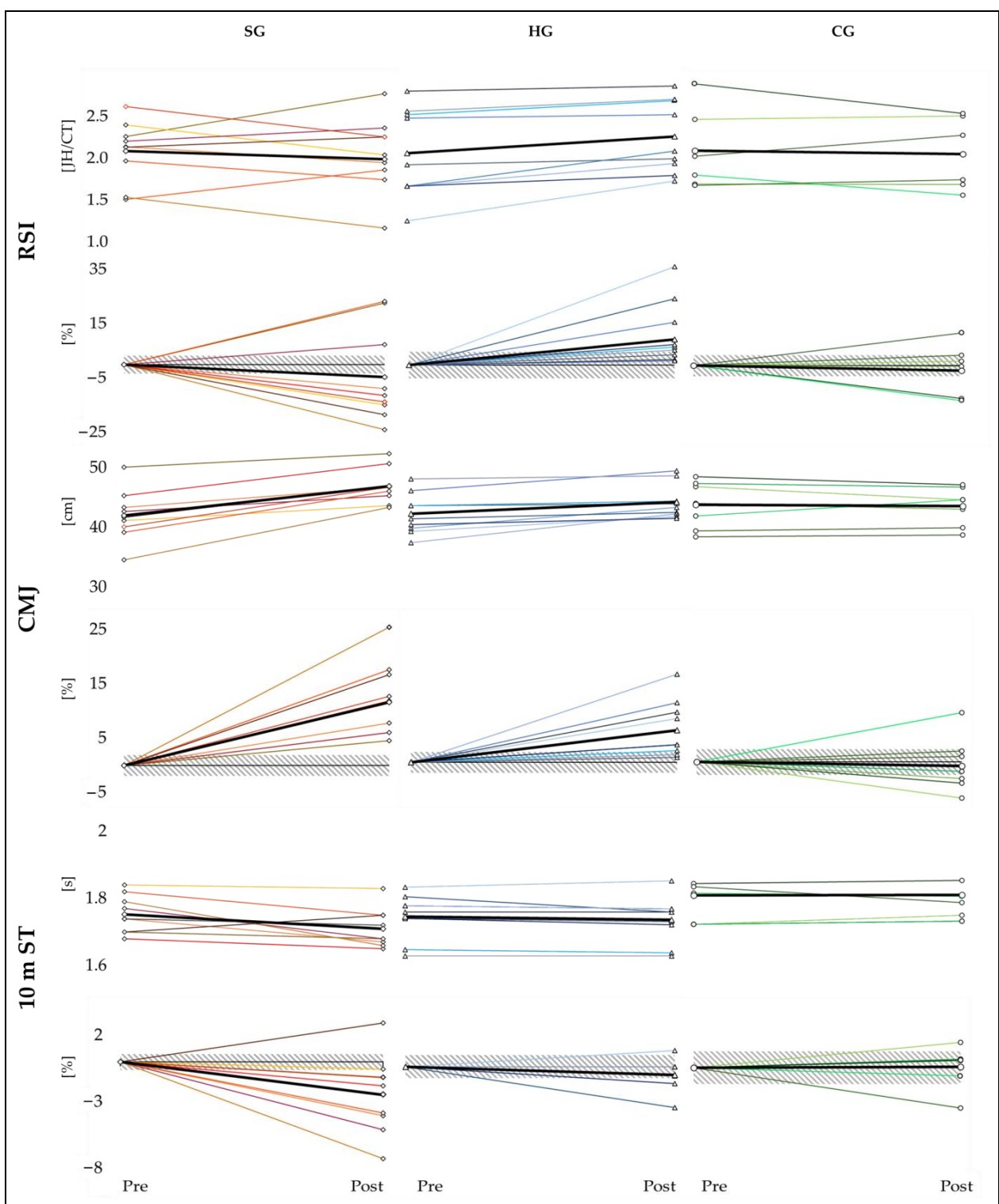

**Figure 1.** Illustration of absolute and percentage changes in selected speed performances of the different intervention groups pre- and post-intervention (the dashed area represents the smallest worthwhile change [SWC] for a small effect). Each color represents one participant (red-yellow color scheme with squares = SG; blue color scheme with triangles = HG; green color scheme with circles = CG).

**Table 3.** Comparison of jumping and speed performance among the SG (*n* = 9), HG (*n* = 9), and CG (*n* = 7) pre- and post-intervention.

| | | Pre | Post | Qualitative Inferences for Effect Magnitude (Mean Difference, ±90% CL) | rANOVA |
|---|---|---|---|---|---|
| RSI | SG | 2.09 ± 0.37 | 1.99 ± 0.45 | # (−0.10, ±0.15) | |
| | HG | 2.05 ± 0.51 | 2.23 ± 0.50 | ## (0.19, ±0.14) | *p* = 0.061 |
| | CG | 2.10 ± 0.42 | 2.06 ± 0.37 | # (0.04, ±0.18) | |
| CMJ [cm] | SG | 42.0 ± 4.3 | 46.9 ± 3.0 a | ### (4.88, ±1.80) | |
| | HG | 42.1 ± 3.3 | 44.0 ± 2.8 | ### (1.89, ±0.98) | *p* < 0.001 |
| | CG | 43.0 ± 4.1 | 42.8 ± 3.3 | (−0.23, ±1.23) | |
| PSJT [cm] | SG | 55.6 ± 6.4 | 59.5 ± 7.4 a | ### (3.93, ±1.45) | |
| | HG | 54.4 ± 5.2 | 55.2 ± 5.2 | # (0.86, ±1.30) | *p* = 0.002 |
| | CG | 55.2 ± 4.8 | 54.9 ± 3.9 | (−0.29, ±1.57) | |
| CH [cm] | SG | 663 ± 0.6 | 698 ± 0.5 | ### (0.35, ±0.16) | |
| | HG | 624 ± 0.5 | 630 ± 0.5 | # (0.06, ±0.19) | *p* = 0.082 |
| | CG | 671 ± 0.4 | 671 ± 0.5 | (0.00, ±0.23) | |
| 5 m ST [s] | SG | 1.02 ± 0.04 | 0.97 ± 0.05 | ### (−0.05, ±0.02) | |
| | HG | 1.03 ± 0.05 | 1.02 ± 0.05 | # (−0.01, ±0.03) | *p* = 0.088 |
| | CG | 1.10 ± 0.08 | 1.10 ± 0.09 | # (0.00, ±0.03) | |
| 10 m ST [s] | SG | 1.75 ± 0.06 | 1.72 ± 0.06 a,b | ### (−0.04, ±0.02) | |
| | HG | 1.78 ± 0.07 | 1.77 ± 0.07 | (−0.01, ±0.02) | *p* = 0.050 |
| | CG | 1.84 ± 0.11 | 1.84 ± 0.11 | (0.00, ±0.03) | |
| 20 m ST [s] | SG | 3.03 ± 0.09 | 2.94 ± 0.11 a,b | ### (−0.09, ±0.03) | |
| | HG | 3.08 ± 0.11 | 3.05 ± 0.10 | # (−0.02, ±0.03) | *p* = 0.002 |
| | CG | 3.12 ± 0.14 | 3.13 ± 0.14 | (0.01, ±0.04) | |

Values are given as mean ± SD; a = sig. dif. to pre, b = sig. dif. SG–CG; chances that the true magnitude of the effects is beneficial, # = possibly (25–75%), ## = likely (75–95%), ### = very likely (95–99.5%); CL = confidence limits, RSI = reactive strength index, CMJ = countermovement jump, PSJT = pivot–step–jump test, CH = crossover triple-hop test, 5 m ST = 5 m sprint time, 10 m ST = 10 m sprint time, 20 m ST = 20 m sprint time.

The results of the magnitude-based inferences support the results of the pairwise comparisons by showing that the likelihood that the true magnitude of the effect is beneficial in TT was highest in the SG and the HG. Based on the small effect, the changes in both the HG and SG are highly likely to be beneficial, and the same applies to the Avg Vel 1a of both the SG and HG, while the change in the CG can only be classified as likely beneficial. For Avg Vel 1b, only the change in the SG was classified as highly likely beneficial, while for Dec Dist, the change in the HG was classified as highly likely and that in the CG as likely beneficial. Statistical values regarding CoD performance are shown in Table 4. Moreover, the individual values, as well as the percentage changes, are illustrated in Figure 2 using Avg Vel 1a, Avg Vel 1b, and Dec Dist as examples.

Correlation analysis revealed significant relationships between the changes in the performance of CMJ to PJST (r = 0.73), CH (r = 0.41), 5 m ST (r = −0.66), 10 m ST (r = −0.47), 20 m ST (r = −0.67), TT (r = −0.48), and Avg Vel 1a (r = 0.50). Moreover, from PSJT to CH (r = 0.63), 5 m ST (r = −0.59), 10 m ST (r = −0.52), 20 m ST (r = −0.67), TT (r = −0.48), and Avg Vel 1a (r = 0.52). We found further correlations between the changes from 5 m ST to 10 m ST (r = 0.70), 20 m ST (r = 0.65), and TT (r = 0.40). In addition, between the change from 10 m ST to 20 m ST (r = 0.81), as well as from 20 m ST to TT (r = 0.55) and Avg Vel 1a (r = −0.51). Additionally, between TT and Avg Vel 1a (r = −0.81). The corresponding correlation matrix can be found in the appendix (Figure A3).

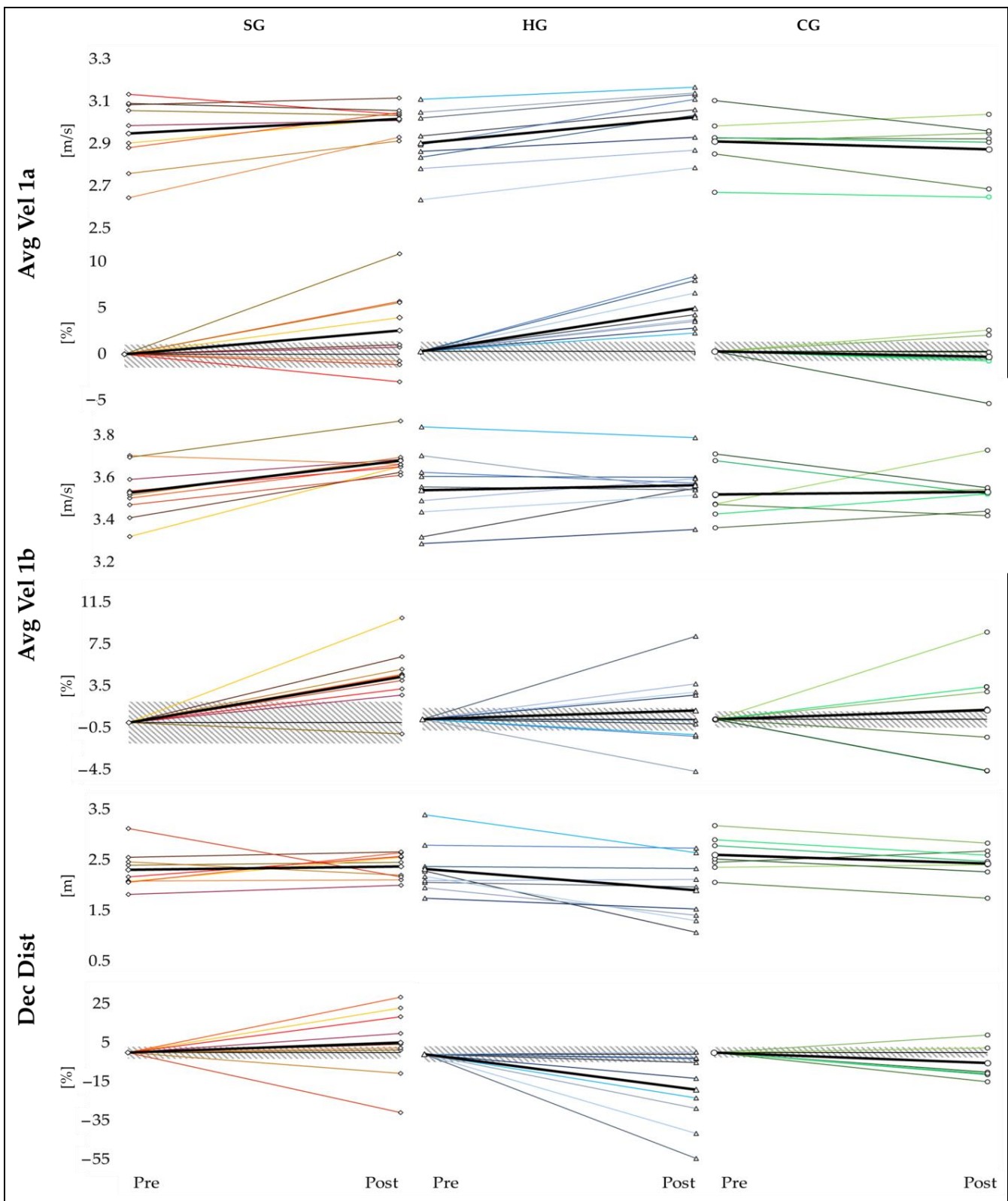

**Figure 2.** Illustration of absolute and percentage changes in selected change of direction (CoD) performances of the different intervention groups pre- and post-intervention (the dashed area represents the SWC for a small effect). Each color represents one participant (red-yellow color scheme with squares = SG; blue color scheme with triangles = HG; green color scheme with circles = CG).

**Table 4.** Comparison of CoD speed performance in the modified 5-0-5 test between SG (*n* = 9), HG (*n* = 9), and CG (*n* = 7) pre- and post-intervention.

| | | Pre | Post | Qualitative Inferences for Effect Magnitude (Mean Difference, ±90% CL) | rANOVA |
|---|---|---|---|---|---|
| Total Time [s] | SG | 2.96 ± 0.14 | 2.89 ± 0.02 a | ### (−0.07, ±0.03) | |
| | HG | 2.97 ± 0.13 | 2.89 ± 0.15 a | ### (−0.08, ±0.03) | *p* = 0.010 |
| | CG | 3.06 ± 0.05 | 3.09 ± 0.13 | (0.03, ±0.03) | |
| Avg Vel 1a [m/s] | SG | 3.00 ± 0.23 | 3.10 ± 0.17 | ### (0.10, ±0.06) | |
| | HG | 2.95 ± 0.16 | 3.08 ± 0.15 a | ### (0.13, ±0.01) | *p* = 0.002 |
| | CG | 2.96 ± 0.14 | 2.88 ± 0.20 | ## (−0.08, ±0.07) | |
| Avg Vel 1b [m/s] | SG | 3.53 ± 0.12 | 3.68 ± 0.10 a,b | ### (0.15, ±0.06) | |
| | HG | 3.54 ± 0.17 | 3.56 ± 0.11 | (0.02, ±0.07) | *p* = 0.036 |
| | CG | 3.52 ± 0.12 | 3.53 ± 0.10 | (0.01, ±0.08) | |
| Dec Dist [m] | SG | 2.31 ± 0.38 | 2.38 ± 0.25 | # (0.07, ±0.02) | |
| | HG | 2.10 ± 0.43 | 1.74 ± 0.52 | ### (−0.36, ±0.20) | *p* = 0.075 |
| | CG | 2.35 ± 0.32 | 2.20 ± 0.30 | ## (−0.15, ±0.26) | |

Values are given as mean ± SD; a = sig. dif. to pre, b = sig. dif. SG–CG; chances that the true magnitude of the effects is beneficial, # = possibly (25–75%), ## = likely (75–95%), ### = very likely (95–99.5%); CL = confidence limits, Avg Vel 1a = average velocity from start to change of direction, Avg Vel 1b = average velocity after change of direction, Dec Dist = distance from the first drop in velocity to the change of direction.

## 4. Discussion

The results of this study revealed that some of the jump and speed performance parameters improved significantly in the SG. Specifically, the SG showed significant improvements in CMJ, PSJT, 10 m ST, and 20 m ST performance from pre- to post-intervention. Moreover, both the HG and the SG showed improvements in CoD performance. However, a more detailed analysis shows that only the HG improved in Avg Vel 1a, while Avg Vel 1b only improved in the SG. These findings suggest that sand could be an effective training surface to improve jumping, sprinting, and CoD performance.

To examine the immediate influence of sand as a training surface, the acute effects of one training session on CMJ, LA, HR, and RPE were compared between the SG and the HG, the findings of which revealed an enhancement in CMJ performance, higher LA levels, and increased RPE values in the SG, indicating both an acute performance enhancement and heightened metabolic demands compared to training on a hard surface. These results align with previous research that has examined the metabolic responses to sand-based training [25,43,44]. The inherent instability and force-absorbing characteristics of sand-based training impose heightened stabilization demands during the support phase of the stride, accompanied by an elongated ground contact time and, thus, a prolonged time under tension of the muscles. Furthermore, the reduced efficiency of force transmission, manifested by the foot sliding during push-off, necessitates greater muscular exertion to achieve the desired movement patterns [25]. Therefore, an immediate enhancement in CMJ performance following sand training could be attributed to the heightened muscle activation induced by the interplay of the aforementioned factors, as this could elicit a potentiation effect. However, the required increase in muscular activation also leads to heightened metabolic demands, as indicated by elevated LA levels and an increased RPE. High-intensity training on sand thus leads to the accelerated depletion of phosphocreatine, consequently heightening the demand for glycogenolysis and glycolysis. If the resulting accumulation of LA becomes excessive, muscle acidosis could result, thus inhibiting contraction processes and decreasing performance [45]. It is generally accepted that training to improve speed must be carried out without a drop in performance over the course of the workout. As such, studies focused on sprinting and jumping have confirmed the importance of sufficient recovery to maintain running speed and jumping performance [46,47]. However, based on the longitudinal improvements observed in performance throughout this study, it can be

inferred that the LA value during training was not sufficiently elevated to hinder or restrict performance. Because this study was conducted with high-level basketball players, LA habituation may have influenced the results, and it should be considered that outcomes could differ among less-trained subjects.

In line with previous studies, the current findings provide evidence supporting the effectiveness of sand training in improving both jumping and sprint performance on a hard surface [27,44,48–50]. This indicates the potential transfer of training effects from sand to hard surfaces, which seems surprising initially, as training to enhance these athletic capabilities typically prioritizes the favorable attributes of firm surfaces that enable the generation of high stretch loads, effective storage of elastic energy, and activation of the stretch reflex. These biomechanical factors play a critical role in facilitating the SSC and the resulting concentric muscle contractions, leading to the refinement of athletic capabilities in terms of jumping and sprinting [51,52]. Conversely, sand training results in a significant dissipation of elastic energy, leading to extended ground contact times and decreased movement efficiency. This causes a greater demand for concentric muscle actions, increases energy costs and the level of muscle activation [23], and challenges the ankle to generate vertical force. Therefore, jumping on sand necessitates a more vigorous concentric push-off phase, which has been demonstrated as an important predictor of jumping performance [53]. Furthermore, all plyometrics performed on sand rely on the long SSC due to extended ground contact times, whereas some performed on hard surfaces rely more on the short SSC. Correspondingly, enhancements in vertical jump performance, in particular, have been observed in exercises involving a longer SSC (CMJ, PSJT), where concentric force development makes a greater contribution. Moreover, the findings of the current study demonstrate an improvement in jumping performance reliant on a rapid SSC (RJ), specifically in the HG. Although this improvement does not reach statistical significance, the magnitude-based inferential analysis indicates a likely beneficial effect here. To provide a general overview, all results are shown graphically in Figure 3. Based on our findings, it is reasonable to propose that both training strategies have the potential to improve jump and sprint performance through two distinct and potentially complementary mechanisms. Hard surfaces provide a heightened training stimulus for improving the SSC, whereas sand training may elicit greater adaptations to the contractile properties of concentric muscle contractions.

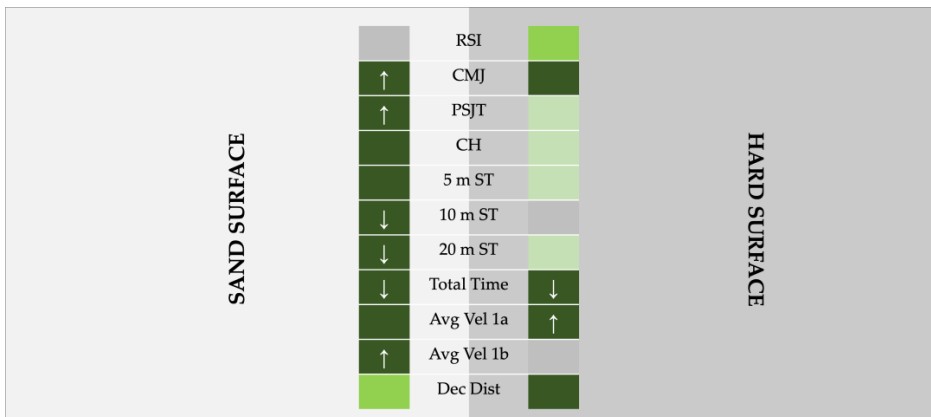

**Figure 3.** Graphical overview of the results (↑ = significant increase; ↓ = significant decrease; light green = possibly beneficial effect; green = likely beneficial effect; dark green = very likely beneficial effect).

The results of the correlation analysis indicate a significant relationship between changes in jumps that rely on the long SSC and sprint performances. The detailed results of the correlation analysis are presented in Figure A3 in the appendix. Accordingly, the results of the present study demonstrate that sand-based training leads to improvements in short sprint performance (10, 20 m), as well. In addition to the highlighted emphasis on concentric force development, the force-absorbing properties of sand impose additional demands,

potentially contributing to the improved sprint performance observed. The diminished energy return to the body on sand induces altered kinematic movement strategies when compared to exercises performed on solid surfaces. Running at higher speeds on hard surfaces is commonly associated with specific kinematic characteristics, such as an increased stride frequency, greater hip flexion, a forward-leaning trunk posture, and enhanced plantar flexion [22,54]. Remarkably, similar features were observed in kinematic studies on individuals running on sand [22]. Consequently, the lower vertical push-off velocity on sand leads to an increased stride frequency. In addition, running on sand induces greater hip flexion, which can be attributed to the adoption of a forward-leaning trunk posture. This postural adjustment shifts the body's center of gravity anteriorly during the stance phase, facilitating the generation of a higher horizontal ground reaction force against the shifting sand surface, while the slippery nature of the sand surface necessitates increased plantar flexion [22]. Hence, the improved sprint performance demonstrated by athletes training on sand may be indicative of a deliberate adaptation to their running technique, thus potentially aiding their overall performance. In addition, the recruitment of supplementary motor units, in conjunction with the discussed enhancements in contractile properties pertaining to concentric force development, is likely to exert a synergistic influence on the amplification of sprint performance. Previous studies have supported these assumptions by demonstrating significant increases in muscle activation and force development following plyometric training on sand [55,56], which is particularly relevant, as the performance of the initial sprint phase heavily relies on the capacity for rapid and high concentric force development [54].

In line with the previous research, the present findings suggest that both sand-based training [27,49] and training on hard surfaces have the potential to enhance CoD performance [28,29,57]. The present study contributes to the existing literature by providing detailed insights into the effects of sand-based training and training on hard surfaces on CoD performance. Further, analyzing the average velocities before and after the CoD maneuver revealed notable differences between the two training conditions. Specifically, the higher velocities observed before the CoD maneuver (Avg Vel 1a), along with the tendency toward a reduction in Dec Dist, suggest a delayed deceleration onset compared to the SG. Although statistical significance in the comparison of Dec Dist was not reached, the magnitude-based inferences strongly suggest a beneficial effect of training on hard surfaces in the reduction in Dec Dist (Figure 3). These findings indicate that training on hard surfaces may enhance an athlete's ability to decelerate effectively, enabling them to maintain higher speeds for a longer duration before executing the CoD maneuver [55]. These observations may be attributed to the biomechanical demands imposed by deceleration on a hard surface, which may enhance the eccentric strength capacity of the muscles to a greater extent than on soft surfaces [23]. The nature of hard surfaces requires greater force absorption and eccentric muscle actions during deceleration, leading to increased loading and adaptation of the muscles involved [22,23,55]. Consequently, athletes training on hard surfaces may experience greater improvements in eccentric strength, which can contribute to their ability to decelerate effectively and maintain higher speeds before initiating a CoD. The delayed deceleration onset observed in the HG implies the potential for improved deceleration capabilities, which could be advantageous in sports requiring rapid CoDs [27,55]. Furthermore, the slight increase in Dec Dist observed in the SG, although not statistically significant, implies a potential adjustment to the deceleration strategy when training on sand surfaces, and this may result in longer deceleration distances compared to training on hard surfaces (Figure 3). Moreover, the correlations between the improvements in the CMJ, the PSJT, and the 20 m ST to those of AVG Vel 1a suggest that improvement in force production derived from the long SSC is also reflected in the performance improvement in this variable (Figure A3). These findings highlight the adaptability of athletes to different training surfaces and the potential for specific adaptations to CoD performance.

In contrast, the higher Avg Vel 1b value observed in the SG indicates superior acceleration compared to the HG, and this discrepancy in velocity can be attributed to the

kinematic and kinetic adaptations induced by sand-based training, which likely contribute to enhanced acceleration performance. Importantly, these improvements may carry over to acceleration performance on hard surfaces, as discussed earlier [22,54–56]. Interestingly, the correlation analyses did not reveal any relationship to other variables (Figure A3). This could indicate that the acceleration after the CoD is more dependent on other factors than the force production generated in the long SSC. Therefore, an additional explanation for the higher Avg Vel 1b value observed in the SG could be attributed to the inherent instability of the sand surface, which may lead to increased activation of the muscles responsible for stabilizing the leg axis and trunk, thereby enhancing the stability and balance of the lumbo-pelvic complex during explosive sports activities. Supporting this assumption, studies have shown an improvement in dynamic and static balance as a result of sand-based training [44,49]. The increased activation of the stabilizing muscles on sand can be explained by the need to counteract the unpredictable and shifting nature of the grains, demanding greater motor control and coordination. This activation may result in improved neuromuscular adaptations, which could enhance the ability to maintain stability and balance during movements, including CoD maneuvers [19]. The ability to stabilize the lumbo-pelvic complex effectively allows athletes to generate and transfer forces more efficiently, resulting in an increased average velocity after the CoD [58]. In addition, athletes with superior balance are likely to possess the ability to reorient themselves more efficiently and initiate acceleration earlier following the CoD maneuver, which is in line with previous research indicating that dynamic balance specifically plays a crucial role in influencing CoD performance [58–60]. The enhanced stability and balance achieved through sand-based training may provide athletes with a competitive advantage in executing CoD maneuvers by allowing for quicker repositioning and faster acceleration [19]. While these explanations are plausible, further research is necessary to substantiate or challenge these hypotheses through comparative studies that explore the effects of speed and plyometric training programs on both sand and solid surfaces. Specifically, investigating the role of balance and stability in relation to CoD performance would provide valuable insights into the underlying mechanisms. Moreover, future studies could examine how different training volumes influence outcome variables, providing clarity on the relationship between training dose and performance outcomes in the context of sand-based training.

*4.1. Limitations*

The relatively small sample size and group heterogeneity in this study suggest that results interpretation should be approached with caution. Given this study's setting in a professional sports context involving multiple teams, randomization of participants to intervention groups was unfeasible, suggesting the possibility of systematic group differences that may potentially impact this study's results and highlighting the importance of careful interpretation. Although the overall training volume, including the number of team training, strength, and conditioning sessions, remained the same, it is worth considering whether different training contents and coaching styles regarding the regular team trainings may have influenced this study's results. As such, further studies are necessary to determine the reproducibility and generalizability of the present results, providing an opportunity to either falsify or verify the findings.

*4.2. Practical Recommendations*

The findings of this study suggest that sand-based training may be advantageous for improving jumps relying on a long SSC and short sprints compared to training on hard surfaces. Both training surfaces seem effective at improving CoD performance, likely through distinct underlying adaptations. However, sand-based training has been shown to be more metabolically demanding. As such, it is suggested that when training on sand with the aim of enhancing speed performance, shorter exercise durations and longer recovery periods should be implemented to optimize performance and minimize muscle acidosis. Moreover, it is likely that sand-based training offers unique benefits by targeting specific

underlying qualities crucial to athletic performance. Due to the inherent instability of sand, engaging in sand-based training leads to increased muscle activation and enhancements in motor unit recruitment, stability, and balance. Furthermore, the force-absorbing nature of sand requires individuals to generate greater muscular effort during the concentric phase of movement to overcome the resistance provided by the shifting sand. This increased demand on concentric muscle actions during sand-based training leads to specific adaptations that prioritize the development of concentric force production capabilities. As a result, there may be a simultaneous reduction in adaptations related to eccentric performance as the training focus shifts toward concentric muscle actions. The same force-absorbing properties of sand reduce any impact on the body's structures when performing high-intensity exercises, such as jumping or speed drills. Considering these factors, incorporating sand-based training in a team sport context is recommended from two perspectives. First, it can be implemented during the initial general preparation phase of a periodized training program in order to prepare the body's structures for the higher subsequent impacts while also targeting the specific qualities mentioned earlier. Sand-based training is particularly useful in this initial phase, as here, variation rather than high levels of specificity is usually favored. This approach helps to adapt the body gradually to subsequently increasing requirements and to reduce the risk of overloading. Second, the reduced impact and increased demands on balance and stabilization make sand-based training beneficial for reintroducing players to specific loads on hard surfaces after recovering from injuries, as it allows for a controlled and progressive transition, promoting rehabilitation and minimizing the risk of re-injury. However, sand-based training cannot replace training on hard surfaces, as training must meet the specific demands of the competition, and most team sports compete on hard surfaces. Therefore, emphasis should be placed on planning a training progression that favors a transition from sand-based to hard surface training, which applies to both post-injury rehabilitation and performance development training.

## 5. Conclusions

The findings suggest that sand-based training elicits specific adaptations that contribute to enhanced athletic capabilities and that it is likely that the improvements observed following sand-based training are due to kinematic and neuromuscular adaptations that may advance running techniques and force development. The inherent instability of sand and its force-absorbing properties lead to greater hip flexion, more torso tilt, and greater plantar flexion, while its instability results in increased muscle activation, enhancing motor unit recruitment, stability, and balance. Furthermore, the extended ground contact times resulting from the force-absorbing nature of sand necessitate greater muscular effort during the concentric phase of a movement, within which the reliance on the long SSC becomes evident. This unique characteristic of sand adds an additional challenge to plyometric exercises performed on sandy surfaces, as the muscles must work harder to overcome the natural overload provided by the constantly shifting grains. Both the increased muscle activation and force absorption in sand-based training contribute to higher metabolic demand and the less efficient utilization of elastic energy. Apparently, both training surfaces have the potential to enhance performance in distinct ways, where sand-based training appears to focus primarily on improving the contractile capacity of the muscles while training on hard ground emphasizes optimizing the utilization of elastic energy. Therefore, it seems reasonable to determine the training surface based on the prioritized training objectives. Regardless, further research is needed to explore optimal training volumes, the role of balance and stability, and the dose–response relationship. In conclusion, sand-based training is a valuable modality for improving jumping, speed, and CoD performance on hard surfaces, with implications for athletes aiming to enhance their athletic abilities.

**Author Contributions:** Conceptualization, J.-L.V. and A.F.; methodology, J.-L.V., A.E. and A.F.; software, J.-L.V. and N.V.; investigation, J.-L.V., J.H., N.B. and M.T.; writing—original draft preparation, J.-L.V., A.E. and A.F.; writing—review and editing, J.-L.V., A.E. and A.F.; visualization, J.-L.V., A.E. and A.F.; supervision, A.F. All authors have read and agreed to the published version of the manuscript.

**Funding:** This research received no external funding.

**Institutional Review Board Statement:** This study was conducted in accordance with the Declaration of Helsinki and approved by the Institutional Ethics Committee of Ruhr University Bochum Faculty of Sport Science (EKS V 11/2022, 29 June 2022).

**Informed Consent Statement:** Informed consent was obtained from all subjects involved in this study.

**Data Availability Statement:** The raw data supporting the conclusion of this article will be made available by the authors on demand, without undue reservation.

**Conflicts of Interest:** The authors declare no conflict of interest.

## Appendix A

| Week | Exercise/Drill | CoD Angle [°] | Distance / Rep [m] | Sets | Reps / Direction | CoD / Sets | CoD / Drill | Distance / Drill | Contacts | Total Distance [m] | Total Contacts |
|---|---|---|---|---|---|---|---|---|---|---|---|
| 1 | CMJ | | | 3 | 5 | | 0 | 0 | 15 | | |
| | 180 - V1a | 180 | 15 | 3 | 1 | 2 | 6 | 45 | | | |
| | Broad Jump | | | 3 | 5 | | | | 15 | | |
| | 180 -V2a | 180 | 15 | 3 | 1 | 2 | 6 | 45 | | 180 | 40 |
| | Lateral Jump | | | 2 | 5 | | 0 | 0 | 10 | | |
| | 90° - V1 | 90 | 20 | 2 | 1 | 3 | 6 | 40 | | | |
| | Curve - V 1a | 90 | 25 | 2 | 1 | 1 | 2 | 50 | | | |
| 2 | CMJ C | | | 3 | 6 | | 0 | 0 | 18 | | |
| | 180 - V1b | 180 | 20 | 3 | 1 | 3 | 9 | 60 | | | |
| | Broad Jump C | | | 3 | 6 | | | | 18 | | |
| | 180 -V2a | 180 | 15 | 3 | 1 | 2 | 6 | 45 | | 195 | 48 |
| | Lateral Jump C | | | 2 | 6 | | 0 | 0 | 12 | | |
| | 90° - V1 | 90 | 20 | 2 | 1 | 3 | 6 | 40 | | | |
| | Curve - V 1a | 90 | 25 | 2 | 1 | 1 | 2 | 50 | | | |
| 3 | CMJ C | | | 3 | 7 | | 0 | 0 | 21 | | |
| | 180 - V1b | 180 | 20 | 3 | 1 | 3 | 9 | 60 | | | |
| | Broad Jump C | | | 3 | 7 | | | | 21 | | |
| | 180 -V2b | 180 | 20 | 3 | 1 | 3 | 9 | 60 | | 210 | 56 |
| | Lateral Jump C | | | 2 | 7 | | 0 | 0 | 14 | | |
| | 90° - V1 | 90 | 20 | 2 | 1 | 3 | 6 | 40 | | | |
| | Curve - V 1b | 90 | 25 | 2 | 1 | 1 | 2 | 50 | | | |
| 4 | SL Step & Jump | | | 3 | 4 | | 0 | 0 | 24 | | |
| | 180 - V1c | 180 | 22 | 3 | 1 | 3 | 9 | 66 | | | |
| | SL Broad Jump A | | | 3 | 3 | | | | 18 | | |
| | 180 -V2b | 180 | 20 | 3 | 1 | 3 | 9 | 60 | | 236 | 66 |
| | SL Lateral Jumps A | | | 3 | 2 | | 0 | 0 | 24 | | |
| | 90° - V2 | 90 | 20 | 3 | 1 | 3 | 9 | 60 | | | |
| | Curve - V 1b | 90 | 25 | 2 | 1 | 1 | 2 | 50 | | | |
| 5 | SL Step & Jump | | | 3 | 4 | | 0 | 0 | 24 | | |
| | 180 - V1c | 180 | 22 | 4 | 1 | 3 | 12 | 88 | | | |
| | SL Broad Jump C | | | 3 | 3 | | | | 18 | | |
| | 90 & 180 - V1a | 180 | 20 | 3 | 1 | 3 | 9 | 60 | | 258 | 78 |
| | SL Lateral Jumps C | | | 3 | 3 | | 0 | 0 | 36 | | |
| | 90° - V2 | 90 | 20 | 3 | 1 | 4 | 12 | 60 | | | |
| | Curve - V 1c | 90 | 25 | 2 | 1 | 1 | 2 | 50 | | | |
| 6 | SL Step & Jump | | | 3 | 4 | | 0 | 0 | 24 | | |
| | 180 - V1c | 180 | 22 | 4 | 1 | 3 | 12 | 88 | | | |
| | SL Broad Jump C | | | 3 | 4 | | | | 24 | | |
| | 90 & 180 - V1a | 180 | 20 | 3 | 1 | 3 | 9 | 60 | | 288 | 84 |
| | SL Lateral Jumps C | | | 3 | 3 | | 0 | 0 | 36 | | |
| | 90° - V1b | 90 | 30 | 3 | 1 | 4 | 12 | 90 | | | |
| | Curve - V 1c | 90 | 25 | 2 | 1 | 1 | 2 | 50 | | | |
| 7 | SL Step & Jump | | | 3 | 4 | | 0 | 0 | 24 | | |
| | 180 - V1c | 180 | 22 | 4 | 1 | 3 | 12 | 88 | | | |
| | SL Broad Jump C | | | 3 | 4 | | | | 24 | | |
| | 90 & 180 - V1b | 180 | 20 | 4 | 1 | 3 | 12 | 80 | | 308 | 96 |
| | SL Lateral Jumps C | | | 3 | 4 | | 0 | 0 | 48 | | |
| | 90° - V1b | 90 | 30 | 3 | 1 | 4 | 12 | 90 | | | |
| | Curve - V 1c | 90 | 25 | 2 | 1 | 1 | 2 | 50 | | | |

| **Warm Up** | |
|---|---|
| Lunges & Sprinter Pose | 6 |
| SLRDL & Sprinter Pose | 6 |
| Lateral Lunges + Rotation | 6 |
| Worlds Greatest | 6 |
| Skippings | 10m |
| A-Skips | 10m |
| Hot Steps | 10m |
| SL Lateral Hops | 5 |
| Ladder Two-One | 1 |
| Ladder Crossover | 1 |
| Ladder lateral In and Out | 1 |

**Figure A1.** Progressive combined plyometric and CoD drill training protocol.

**Appendix B**

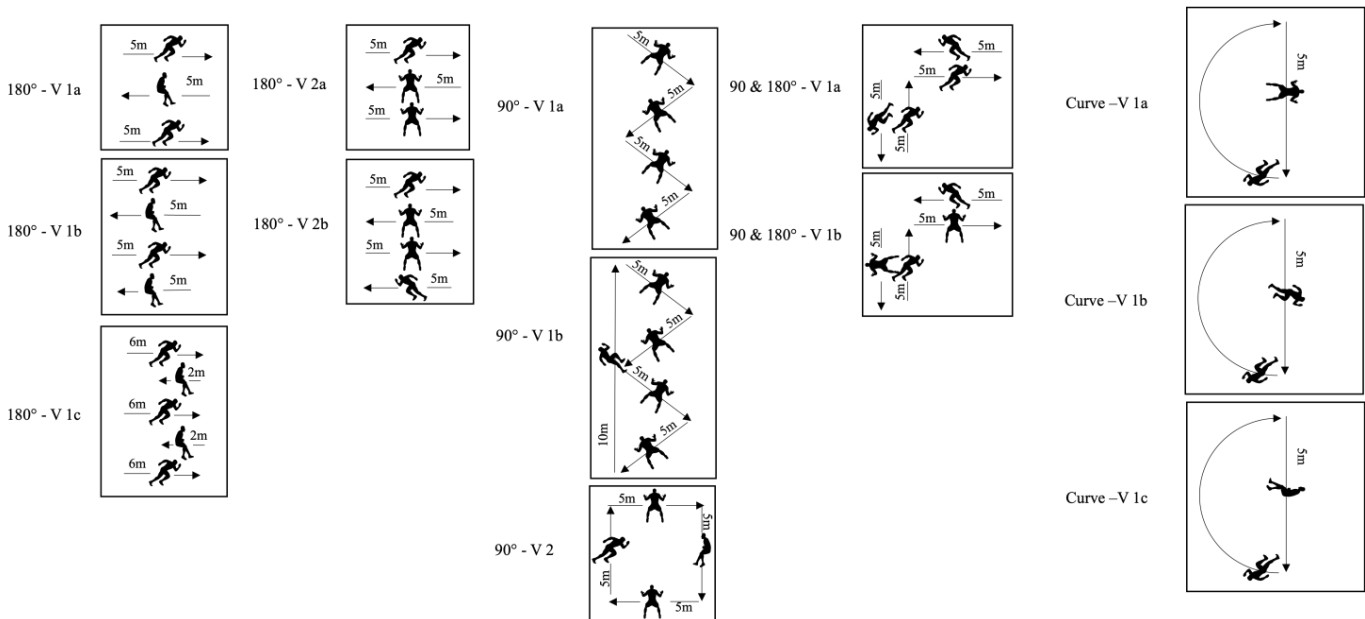

**Figure A2.** Illustration of the various CoD drills.

| | | RSI_Dif | CMJ_Dif | PSJT_Dif | CH_Dif | 5 m ST_Dif | 10 m ST_Dif | 20 m ST_Dif | TT_Dif | Avg Vel 1a_Dif | Avg Vel 1b_Dif | DecDist_Dif |
|---|---|---|---|---|---|---|---|---|---|---|---|---|
| RSI_Dif | Pearson's r | | | | | | | | | | | |
| | df | | | | | | | | | | | |
| | p-Wert | | | | | | | | | | | |
| CMJ_Dif | Pearson's r | −0.261 | | | | | | | | | | |
| | df | 23 | | | | | | | | | | |
| | p-Wert | 0.208 | | | | | | | | | | |
| PSJT_Dif | Pearson's r | −0.201 | 0.727 *** | | | | | | | | | |
| | df | 23 | 23 | | | | | | | | | |
| | p-Wert | 0.335 | <.001 | | | | | | | | | |
| CH_Dif | Pearson's r | −0.299 | 0.410 * | 0.633 *** | | | | | | | | |
| | df | 23 | 23 | 23 | | | | | | | | |
| | p-Wert | 0.147 | 0.042 | <.001 | | | | | | | | |
| 5 m ST_Dif | Pearson's r | 0.351 | −0.663 *** | −0.592 ** | −0.336 | | | | | | | |
| | df | 23 | 23 | 23 | 23 | | | | | | | |
| | p-Wert | 0.086 | <.001 | 0.002 | 0.101 | | | | | | | |
| 10 m ST_Dif | Pearson's r | 0.001 | −0.475 * | −0.515 ** | −0.213 | 0.702 *** | | | | | | |
| | df | 23 | 23 | 23 | 23 | 23 | | | | | | |
| | p-Wert | 0.995 | 0.017 | 0.008 | 0.306 | <.001 | | | | | | |
| 20 m ST_Dif | Pearson's r | 0.049 | −0.613 ** | −0.666 *** | −0.342 | 0.649 *** | 0.807 *** | | | | | |
| | df | 23 | 23 | 23 | 23 | 23 | 23 | | | | | |
| | p-Wert | 0.816 | 0.001 | <.001 | 0.095 | <.001 | <.001 | | | | | |
| TT_Dif | Pearson's r | −0.204 | −0.522 ** | −0.480 * | −0.166 | 0.404 * | 0.360 | 0.553 ** | | | | |
| | df | 23 | 23 | 23 | 23 | 23 | 23 | 23 | | | | |
| | p-Wert | 0.328 | 0.007 | 0.015 | 0.427 | 0.045 | 0.077 | 0.004 | | | | |
| Avg Vel 1a_Dif | Pearson's r | 0.209 | 0.504 * | 0.518 ** | 0.312 | −0.375 | −0.367 | −0.509 ** | −0.806 *** | | | |
| | df | 23 | 23 | 23 | 23 | 23 | 23 | 23 | 23 | | | |
| | p-Wert | 0.317 | 0.010 | 0.008 | 0.129 | 0.065 | 0.071 | 0.009 | <.001 | | | |
| Avg Vel 1b_Dif | Pearson's r | −0.157 | 0.352 | 0.213 | 0.294 | −0.201 | −0.119 | −0.254 | −0.289 | 0.272 | | |
| | df | 23 | 23 | 23 | 23 | 23 | 23 | 23 | 23 | 23 | | |
| | p-Wert | 0.455 | 0.085 | 0.307 | 0.153 | 0.336 | 0.572 | 0.220 | 0.162 | 0.188 | | |
| DecDist_Dif | Pearson's r | −0.013 | 0.245 | 0.198 | 0.121 | −0.167 | −0.143 | −0.141 | 0.046 | 0.116 | 0.221 | |
| | df | 23 | 23 | 23 | 23 | 23 | 23 | 23 | 23 | 23 | 23 | |
| | p-Wert | 0.949 | 0.238 | 0.342 | 0.563 | 0.426 | 0.496 | 0.500 | 0.828 | 0.582 | 0.288 | |

* p < 0.05, ** p < 0.01, *** p < 0.001

**Figure A3.** Correlation matrix of the changes in performance.

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
