# Peer review of "Training on Sand or Parquet: Impact of Pre-Season Training on Jumping, Sprinting, and Change of Direction Performance in Professional Basketball Players"

_applsci, doi:10.3390/app13148518_

Round 1

Reviewer 1 Report

Dear Authors

Thank you for giving me the opportunity to review your prominent research. The analyzed manuscript focuses on assessing and comparing the effects of a combined training protocol consisting of jump exercises and combined change of direction speed drills on a sand surface versus a hard surface, specifically evaluating the combined change of direction performance alongside jump and sprint capabilities.

The manuscript with the reference "applsci-2486885" shows great potential in terms of originality and empirical application. However, there are some adjustments that can be made to make it even stronger, as outlined below. The critical points presented in this review are intended to help you enhance the impact and clarity of your work. Addressing these points will significantly contribute to realizing the full potential of your research. I look forward to seeing the revisions you make to your manuscript.

1. Firstly, it is not recommended to use acronyms or abbreviations in the title, as they may cause confusion. In this case, the abbreviation used is not common and may be confused with another acronym, "COD," which typically stands for "Chemical Oxygen Demand." I suggest using the full term "Change of Direction" instead.

2. I noticed minor grammar, spelling, and syntax errors in the manuscript that may hinder its readability and overall impact. To ensure that the manuscript is error-free and facilitates its dissemination to a wider audience, I strongly recommend considering the assistance of a proofreader.

3. The manuscript contains several lengthy paragraphs that lack proper integration of relevant literature to support the assertions made. To strengthen the validity of the arguments and ensure a more comprehensive discussion, it is crucial to enhance the presentation of supporting references.

4. The abstract is well-structured and provides the necessary information to comprehend the study. However, the conclusion could benefit from reorganization to improve it is precision and conciseness.

5. I would like to seek clarification on the plyometric training performed on the sand. The authors reported between lines 56 and 60: “Several studies have already demonstrated a positive effect on the speed performance of plyometric training on sand [19,23,25,26], and to the best of our knowledge, no previous studies have investigated the effects of combining CoD speed training specifically with plyometrics on a sand surface.” This statement was linearly associated with the purpose of the study. In this context, did you perform plyometric training in the sand? If yes, please explain this.

5.1. In addition, the ground contact time is an important factor to consider as it directly influences the efficiency and effectiveness of plyometric exercises/training. Plyometric activities can be separated into two categories depending upon the duration of the ground contact time: fast plyometric movements (≤ 250 milliseconds (ms) and slow plyometric activities (≥ 251 ms). In this sense, the groups performed different physical activities even if it is plyometric training. Please review this rationale and discuss this information in the paper, especially in the discussion section and, if necessary, in the Material and Methods.

5.2. Still analyzing this point, it was reported between lines 20 and 22 and between lines 541 and 544: "[...] findings suggest that sand-based training [...]". But, in the Material and Methods (lines 99 and 100), was reported: “The training protocol consisted of various CoD exercises in combination with different plyometrics”. Please reorganize this information, as the type of contact surface possibly may have modified the training method. Please reorganize this information, as the type of contact surface may have modified the training characteristics.

6. Since the total sample size consisted of 25 professional basketball players from three different teams, it is important to consider the individual physical training background and its potential influence on the results. Please provide your perspective on this matter and include information regarding the participants' level of training, considering the varied sports preparation models among the three teams.

7. There seems to be a discrepancy in the equipment model name. According to lines 129 and 130, the equipment used for lactate analysis is referred to as "Biosen S-Line 129 sport glucose analyzer (EKF-Diagnostik 141 GmbH, Magdeburg, Germany)”. However, the international model for this equipment is commonly known as "EKF Biosen S-Line Lab+". Please verify and clarify this information.

8. The tables in the manuscript do not adhere to the submission norms. Some tables have missing lines, while others contain unexplained "#" symbols. It is important to ensure that the tables are properly formatted and presented according to the guidelines.

9. Between lines 519 and 522, it is stated that sand-based training leads to specific adaptations that prioritize the development of concentric force production capabilities. Please provide the rationale for this statement and cite the relevant literature that supports it.

In conclusion, I commend you on your valuable work. Addressing the points raised in this review will enhance the overall clarity and coherence of the manuscript, providing readers with a comprehensive understanding of the study's outcomes. This, in turn, will strengthen the quality and credibility of the review.

The manuscript presents several grammar and syntax errors that may hinder the readability and overall impact of the research. As such, I would strongly recommend that the authors consider a proofreader to ensure that the manuscript is polished and error-free.

Author Response

Dear reviewer,

first, we would like to thank you for your constructive comments and let you know that we appreciate the time and expertise you have invested in improving our manuscript. All changes to the manuscript are highlighted in yellow to make the adjustments easier to follow.

Reviewer 1

  1. Thank you for bringing this point to our attention. As suggested, we have now placed the whole term 'change of direction' in the heading.
  2. This point surprised us because we had the manuscript commercially proofread before we submitted it. However, we will of course have it proofread again for the accuracy of the English language.
  3. With regard to the references, we have once again completely screened the discussion and have added references where this appeared to be beneficial.
  4. We agree that a slightly more detailed conclusion in the abstract would be helpful. Therefore, we have clarified the conclusion as follows: `As such, the findings suggest that sand-based training elicits kinematic adaptations, increased muscle activation and a shift towards the concentric force development that all contribute to enhanced athletic capabilities.´
  5. To be clearer, we have replaced the term jump exercises in line 63 with plyometric exercises. In addition, we have checked the entire manuscript for consistent use of the term. Furthermore, the detailed training protocol can be found in the appendix.
    • Much appreciation for this comment. Following your suggestion, we have included the following information in the discussion: `Conversely, sand training results in a significant dissipation of elastic energy, leading to extended ground contact times and a decreased movement efficiency. This causes a greater demand for concentric muscle actions, increases energy costs, the level of muscle activation [23], and challenges the ankle to generate vertical force. Therefore, jumping on sand necessitates a more vigorous concentric push-off phase, which has been demonstrated an important predictor of jumping performance [53]. Furthermore, all plyometrics performed on sand rely on the long SSC due to extended ground contact times, whereas some performed on hard surfaces rely more on the short SSC.´ We have also added the following in the conclusion: `Furthermore, the extended ground contact times resulting from the force-absorbing nature of sand necessitate greater muscular effort during the concentric phase of movement, within which the reliance on the long SSC becomes evident. This unique characteristic of sand adds an additional challenge to plyometric exercises performed on sandy surfaces, as the muscles must work harder to overcome the natural overload provided by the constantly shifting grains.´
    • We agree with you on this point and have added the following sentence to the methods:` The protocol for both intervention groups was identical, but all jumps performed on sand rely on the long stretch shortening cycle (SSC) due to the prolonged ground contact times caused by the properties of sand, whereas some jumps rely on the short SSC when performed on hard surfaces.´
  6. As you suggested we have included information regarding the participants level of physical training in the methods section. Since the participants were all highly trained elite level athletes who all compete in the same league, we do not think that differences in training status affected these results. Of course, there were individual (position-specific) differences. However, the performance of the initial diagnostics was comparable for all teams. To clarify this for the reader we added this information in the results section.
  7. We changed the model name of the equipment according to your suggestion.
  8. Thank you for pointing out this issue to us. Apparently the information was lost in the editing process. We have added the information and adjusted the format accordingly.
  9. In the practical recommendations, we only mention points that have been discussed in detail in the discussion section, mentioning the relevant literature. For better readability, we have decided not to mention the references again here.

Reviewer 2 Report

Concerning the manuscript: Training on sand or on parquet: Impact of pre-season training on jumping,

sprinting, and CoD performance in professional basketball players, submitted to Appl. Sci. This very interesting manuscript compares sand and hard surface in the context of training for improving jumping,

sprinting, and CoD performance. This is an interesting study with new information which can be very useful. Overall, the paper will contribute to knowledge and is worthy of publication. With the utmost respect, allow me to give you a few suggestions.

Introduction

·  The introduction is well written, but the novelty could be more highlighted. The authors do not explain the reason why they chose to analyze training on sand (why not in water or tied conditions? Is exercise on sand surface physiological more demanding? and in which way it could relate with jump and sprint performances. I would like to know the authors' line of reasoning about this. Thoughts described in line 351 could already appear in the introduction.

Methods and results

·  Congratulations for the quality of the methods (description of mathematical for example).

·  Give details about the procedures. It is important to describe when data were collected (time of day, environmental condition, season of data collection, data collection duration). Were researchers trained or had training in conducting tests?

·  Information on symbol “#” must be included (table 3 and 4).

·  The authors must insert more mathematical details about smallest worthwhile change (SWC) in the Statistical section. Give a fictitious example.

·  Regarding data exploration, I suggest the inclusion of Pearson’s correlations for understanding physiological interactions. This could be made using data from participants of all groups (for amplifying the sample amount). If using figures, I suggest that you accurately discriminate the groups using different symbols or colors. For example:

SG - > square red,

HG- > triangle green,

CG - > circle blue,

·  In figure 1 and 2, I suggest that you accurately discriminate the participants using different colors.

•The authors should report more details about Avg Vel 1a and Avg Vel 1b in method sections.

·  It would be interesting to apply the trapezoidal method to determine the area under the curve in variables collated over a time period. This could be added in a supplementary file.

Discussion

•The authors should make a recommendation about the use sand surface in the context of training periodization.

·  Didactics would improve with the inclusion of a figure in the discussion (some scheme drawn by the authors) listing the changed variables by sand vs hard surface. An example below:

Author Response

Dear reviewer,

first, we would like to thank you for your constructive comments and let you know that we appreciate the time and expertise you have invested in improving our manuscript. All changes to the manuscript are highlighted in yellow to make the adjustments easier to follow.

  1. Thanks for the comment. we have tried to make the line of reasoning and the novelty of the article clearer in the introduction. For this we have added the following: `The instability and force-absorbing properties of sand require heightened stabilization, elongated ground contact time, and increased muscular exertion [44]. This makes sand training a potentially valuable tool for improving speed, especially CoD speed. Since speed and CoD training is usually associated with high loads on the musculoskeletal system, sand-based training might be a way to achieve this with reduced impact.´
  2. Following your suggestion, we have included in the measurements section the proposed information as follows: ` The teams' performance assessments took place in their training facilities, with all participants being tested on parquet surfaces. These assessments were consistently scheduled in the early evening. Since the tests were conducted indoors in facilities that meet the league's standards, the environmental conditions were the identical. The test battery, which required two hours to complete, was administered exclusively by trained test administrators.´
  3. Thank you for pointing out this issue to us. Apparently the information was lost in the editing process. We have added the information and adjusted the format accordingly.
  4. As suggested, we have indicated how the smallest worthwhile change is calculated and illustrated the calculation with an example. The added passage reads as follows: ` To calculate this, the between subject standard deviation of each performance variable was multiplied by 0.2 (. For instance, if an athlete jumps 45 cm in the CMJ with a standard deviation of 4 for this test in their population, the athlete would have to jump 0.8 m higher to achieve a meaningful change.´
  5. Thank you for this interesting idea. As proposed, we have calculated a correlation analysis with the total sample. We have described this in the methods section as well as in the results and have also included it in the discussion. We believe that these results add valuable information to the discussion on sprint performance improvements as well as CoD performance.
  6. With regard to the figures, we have modified them according to your suggestion, so that the groups are now represented by different symbols (squares, triangles and circles) and color schemes. In addition, the individual participants can be distinguished by different colors, which are identical throughout.
  7. Following the suggestion, we have added information about the average velocity before and after the CoD maneuver in the description of the modified 5-0-5 test. We have also mentioned the other variables at this point. The added sentence reads as follows: `The device was used to measure the total time required, average velocity before (Avg Vel 1a) and after the CoD manoeuvre (Avg Vel 1b) as well as the deceleration distance (Dec Dist), starting with the first decrease in velocity.´
  8. This is an interesting point. Since such an analysis of the data would exceed the time frame of the revision, we have decided against including this supplementary point. As stated in the data availability statement, we are open to provide the data upon request.
  9. Following your advice, we have pointed out the use of sand training in the context of long-term periodization in the practical recommendations section. The corresponding sentence reads as follows: ` First, it can be implemented during the initial general preparation phase of a periodized training program in order to prepare the body’s structures for the higher subsequent impacts, while also targeting the specific qualities mentioned earlier. Sand-based training is particularly useful in this initial phase, as here usually variation rather than high levels of specificity is favored, in order to prepare the body for the increasing demands of more specific training. This approach helps to adapt the body gradually to subsequently increasing requirements and to reduce the risk of overloading.´
  10. Again, an excellent proposal that we gladly followed, although our solution is a little different from your suggestion. We agree that a graphical overview of the results contributes to enhance the quality of the manuscript.

Round 2

Reviewer 1 Report

Congratulations to the authors for the great work.

I have no further requests or comments.

English-specific errors, such as a missing comma before the "and", were detected. Minor editing of the English language is required.

Author Response

Thank you for your effort.